# Engineering Plastic Eating Enzymes Using Structural Biology

**DOI:** 10.3390/biom13091407

**Published:** 2023-09-19

**Authors:** Amelia Barclay, K. Ravi Acharya

**Affiliations:** Department of Life Sciences, University of Bath, Claverton Down, Bath BA2 7AY, UK; ameliabarclay26@gmail.com

**Keywords:** plastics, polyethylene terephthalate (PET), PETase, protein engineering, structural biology, hydrophobicity, crystallinity, thermostability, industrial challenges

## Abstract

Plastic pollution has emerged as a significant environmental concern in recent years and has prompted the exploration of innovative biotechnological solutions to mitigate plastic’s negative impact. The discovery of enzymes capable of degrading specific types of plastics holds promise as a potential solution. However, challenges with efficiency, industrial scalability, and the diverse range of the plastic waste in question, have hindered their widespread application. Structural biology provides valuable insights into the intricate interactions between enzymes and plastic materials at an atomic level, and a deeper understanding of their underlying mechanisms is essential to harness their potential to address the mounting plastic waste crisis. This review article examines the current biochemical and biophysical methods that may facilitate the development of enzymes capable of degrading polyethylene terephthalate (PET), one of the most extensively used plastics. It also discusses the challenges that must be addressed before substantial advancements can be achieved in using these enzymes as a solution to the plastic pollution problem.

## 1. Introduction

Plastics have become an integral part of modern life due to their versatility and low production costs. Conventional plastics are non-biodegradable which makes them a useful, durable material; however, in recent years the amount of plastic waste has become a mounting environmental concern. Their prolonged existence in the environment, notably marine ecosystems, is disrupting the delicate balance of the natural world and more recently the potential threat of microplastics to human health has been identified [1].

Plastics are polymers composed of repeating units of small organic molecules with the majority manufactured from non-renewable fossil fuels. Their persistence within the environment is primarily attributed to their hydrophobic properties and the absence of hydrolysable functional groups in numerous plastics, including polyethylene (PE), polystyrene (PS) and polyvinyl chloride (PVC). The polymer polyethylene terephthalate (PET) is one of the most prevalent plastics and has become a major contributor to plastic pollution; however, it has a susceptibility to degradation as it contains hydrolysable ester bonds [2].

Evidence that enzymes can degrade polyesters dates back more than four decades [3]. Since then, researchers have identified various organisms including bacteria, fungi, and insects capable of degrading specific plastics. Esterases have emerged as promising candidates for plastic degradation. Among these, cutinases stand out as a subtype of esterase capable of cleaving the ester bonds within the cutin polymer, a constituent of plant waxy cuticles. Cutin is a complex polyester composed of ester bonds formed between hydroxy and hydroxyepoxy fatty acids which give it a significant hydrophobic nature [4]. Consequently, it is not surprising that cutinases are often recognised as the most efficient for plastic degradation, possessing a natural affinity for hydrophobic plastics and an ability to hydrolyse the ester bonds found in PET. Through detailed structural investigations, it is understood that cutinases belong to the α/β hydrolases superfamily and possess the Ser-Asp-His catalytic triad. The potential interactions between the enzymes and plastic substrates have also been identified and this knowledge leveraged to engineer improved variants of cutinases with enhanced plastic degradation capabilities [5]. However, these enzymes still face challenges in terms of efficiency, largely attributed to the presence of repeating aromatic terephthalate units that elevate the crystallinity of PET. As a result, the degradation process by these enzymes is slow and often incomplete.

In 2016, a research team isolated the bacterium *Ideonella sakaiensis* from a recycling plant which secretes two enzymes capable of degrading PET [6]. The first enzyme, called PETase, breaks down PET into mono (2-hydroxyethyl) terephthalic acid (MHET). This is then broken down further by MHETase, which produces terephthalate (TPA) and ethylene glycol (EG) and these byproducts can then be metabolised by other bacteria as a carbon source (Figure 1). This discovery yielded much excitement due to the fact this enzyme could survive on only PET as a food source indicating the bacterium has specifically evolved to the breakdown PET rather than just re-directing its existing cutinise activity.

The excitement surrounding the degradation of PET through the subsequent actions of PETase and MHETase has resulted in a surge of research publications detailing the three-dimensional crystal structures of these enzymes, offering valuable insights into their catalytic mechanisms. Similar to the cutinases, PETase employs a tunnel-shaped active site capable of accommodating PET. The catalytic triad of the enzyme has been identified [8], prompting further investigations into enhancing its catalytic efficiency [9] and thermal stability [10] using rational design and directed evolution. The structural knowledge gained from these studies has guided the selection of mutations aimed at improving enzyme-substrate interactions thereby leading to enhanced PET degradation capabilities.

MHETase, the second enzyme in the pair responsible for PET degradation in *I. sakaiensis*, shares an α/β-hydrolase fold with PETase. However, crystal structures of MHETase bound to a non-hydrolysable substrate have revealed an additional lid domain that confers substrate specificity, reminiscent of a feruloyl esterase. Despite this, no residues within the binding pocket are conserved with feruloyl esterases, resulting in MHETase possessing a distinctive binding pocket tailored to accommodate the MHET substrate [11]. There is strong anticipation that guided structural modifications to MHETase will complement PETase, culminating in the creation of a deployable plastic-degrading production line.

The escalating issue of plastic waste has reached a magnitude where traditional methods of plastic waste management, including incineration and landfilling, can no longer effectively address the problem. The concept of a circular economy has been perpetuated by governments and campaigners which has motivated scientists to innovate sustainable solutions to plastic management. Among these, enzyme-enabled approaches provide a promising, more sustainable avenue by breaking down plastics into their constituent molecules, which can then be recycled or degraded further through natural processes. This review will focus on the structural understanding of PETase and how this has led to advancements in engineering this enzyme to enhance its catalytic activity, specificity, and stability.

## 2. Molecular Features of Plastic-Degrading Enzymes

### 2.1. The PETase Catalytic Mechanism

#### 2.1.1. Comparison of PETase with the Homologous Cutinases

Extensive research has been conducted to investigate the molecular mechanisms of these enzymes from *I. sakaiensis*. One study reported a crystal structure of PETase at an unprecedented high resolution of 0.92 Å which allowed for extensive detailed structural comparisons between PETase and the cutinases which share considerable sequence homology [8]. PETase retains the α/β-hydrolase structural fold present in many hydrolase enzymes, consisting of eight β-strands and six α helices. Like cutinases, PETase has a long substrate binding cleft (~40 Å) that is mainly hydrophobic to accommodate the PET polymer. However, a key difference is the notable widening of the substrate binding cleft in PETase compared to cutinases (Figure 2). Austin et al. [8] hypothesized that this widening was an evolutionary adaptation of PETase to accommodate the semi-aromatic and more crystalline PET substrate. To test this hypothesis, a Ser238Phe mutant of PETase was created which surprisingly increased the activity of the enzyme despite the mutant crystal structure reverting back to having a narrower substrate binding cleft. This research firstly highlighted the limitations to our knowledge on how structure corresponds to activity. Secondly, it suggested that additional protein engineering may be able to further increase PETase activity for optimal degradation of plastic waste.

#### 2.1.2. Efforts to Understand the Enzyme-Substrate Complex

Despite the technological advancements in structural biology over the last fifty years, capturing an enzyme-substrate complex, useful for analysing the catalytic mechanism, is still challenging. Due to the rapid hydrolysis by PETase, many groups attempted co-crystallization of PETase with various non-hydrolysable analogue of PET without success [8,12] Others created PETase-trapping mutants [13] or inactive mutants [10], yet this can disturb the delicate network of chemical and physical forces that create the enzyme’s activity, thus hindering our understanding of true mechanistic features.

At the same time, the 21st century has witnessed the increasing application of computation in various scientific disciplines, including structural biology. While the emergence of Artificial Intelligence (AI) prediction tools, such as Alphafold2 and RoseTTAfold, has been met with both anticipation and criticism for their accuracy [14], computation is widely utilised in other areas, such as molecular docking which can model atomic-level interactions between molecules. Important to note is that these simulations are based on rich information obtained from experimentally determined structures in the world repository of protein structures, the Protein Data Bank (PDB), which serves as the foundation for these AI models.

To elucidate the catalytic mechanism of the PETase, a recent study performed molecular docking simulations between the PETase and 2-HE(MHET)4, a molecule consisting of four MHET units that represent PET [12]. The simulations identified two subsites for ligand binding: Subsite I and Subsite II. Subsite I contain a catalytic triad consisting of Ser160, His237, and Asp206, which is typical of a hydrolase reaction, also shared by cutinases, where Ser160 serves as the nucleophilic residue that attacks the carbonyl carbon in the PET ester bond. Surrounding this site is an oxyanion hole created by two nitrogen atoms of residues Tyr87 and Met160, which stabilizes the tetrahedral intermediate. Subsite II is responsible for binding the three MHET-repeats mainly through hydrophobic interactions (Figure 3).

#### 2.1.3. Identification of Unique Enzyme Feature—Wobbling Trp

A Trp185 residue can be located from the PETase crystal structure in Subsite I, proximal to the catalytic triad [13]. This was expected as this Trp185 residue is strictly conserved amongst homologous enzymes. However, in the PETase crystal structure, the researchers found that Trp185 was not well fixed within the structure, instead showing three distinct conformations (Figure 3). Comparison with several available homologous structures showed that the Trp residue usually adopts a single conformation as a C conformer [15]; therefore, the multiple conformations of Trp185 within PETase may be a feature unique to this enzyme. It is hypothesised that this “wobbling” of Trp185 may accommodate the bulkier PET substrate, providing π-stacking forces to stabilize the semi-aromatic PET substrate. Ser214 was identified as a unique residue in PETase, distinct from other homologous structures which possess a histidine in an equivalent position. This indicates that Ser214 is responsible for accommodating the range of conformations of Trp185, which the larger His residue would not permit [16]. To test this, a Ser214His mutant was created, which considerably reduced the activity of PETase but did not completely abrogate it. This suggests that Trp185 “wobbling” is necessary for PET hydrolysis, but other factors within the enzyme must also contribute, warranting further investigation.

### 2.2. Engineering the Enzyme to Enhance Efficiency

The structural investigations on PETase summarised above provide a comprehensive understanding of how these newly discovered enzymes may break down PET. However, in 2018, the United Nations estimated that annual global plastic production exceeded 400 million tonnes, with approximately 85% ending up in the landfill or unmanaged, meaning that it was left as uncollected litter or disregarded in unregulated sites on land or in water [17]. PETase represents the best-studied case of enzymatic degradation, yet can this structural knowledge be used to deploy this and other enzymes to deal with the ever-growing plastic waste that is accumulated each year?

#### 2.2.1. Developing a Thermostable Enzyme to Enhance Efficiency

The application and scaling up of these enzymes have presented challenges due to their relatively slow reaction rates compared to the large amounts of plastic that need to be degraded on an industrial scale. However, it has been extensively documented that raising the temperature to the PET glass transition temperature (>65 °C) reduces the crystallinity of PET, allowing the enzymatic targeting of amorphous PET and thereby enhancing the rate of the hydrolysis process [18]. Since PETase currently operates within a narrow pH range [19] and is not optimal at the glass transition temperature [10], significant research efforts are currently focused on the formidable task of developing a highly thermally stable PETase.

Rational engineering of these PET-degrading enzymes is being pursued with guidance from their three-dimensional molecular structures. It has been established previously that divalent ions have the capacity to increase the thermal stability of some hydrolases [20]. A comprehensive study identified a thermophilic cutinase from a leaf branch compost metagenome, referred to as LCC, which outperformed other PET-degrading enzymes, including PETase, with an optimal temperature of 65 °C. This presented an excellent foundation for the study which went on to enhance the thermostability even further. They identified a divalent metal binding site within the LCC and demonstrated that the addition of calcium ions can stabilize the protein [10]. However, adding ions to reactions can complicate purification at an industrial scale. To avoid this issue, the divalent metal binding site was replaced with a disulphide bridge by creating an Asp238Cys/Ser283Cys mutant. This increased the thermal stability of the enzymes a *Tm* of 72 °C without depending on the addition of calcium but resulted in a 28% decrease in activity. To address this reduction in activity, molecular docking simulations were performed to identify sites for targeted mutagenesis. Combinations of these mutations were experimentally validated, and the resulting Phe243Ile/Asp238Cys/Ser283Cys (ICC) mutant was shown to restore catalytic activity to 122% of wild-type activity. A crystal structure of the ICC mutant reveals that Ile243 is located near the substrate-binding tunnel. This suggests that the mutant’s increased activity is due to an increased PET binding capacity [16]. The ICC PETase mutant is now being used as a starting point for further studies aimed at generating higher thermostable variants through directed evolution [21]. This raises an important point about how far conscious structure-guided enzyme design can take us before we accept that directed evolution may be a more cost-effective way to produce better PETase variants.

One exciting research effort is the discovery of new enzymes, particularly sourced for naturally thermotolerant organisms. These enzymes not only further our understanding of how to create thermostable variants but also provide a better starting point before making structure-guided changes. In one study a total of 37 enzymes, originating from diverse phylogenetic groups, were identified [22]. These enzymes exhibited the capability to degrade amorphous PET film under conditions ranging from pH 4.5–9.0 and temperatures of up to 70 °C. Versatility within the enzymes may also prove useful as there are many different industrial conditions for plastic management and environments with different pH conditions. Discoveries of novel enzymes can now be deposited in the Plastics Biodegradation Database established in 2022 which currently details over 200 enzymes with known ability to the breakdown plastics [23].

#### 2.2.2. Industrial Applications and Challenges

Enzymes are widely used in a range of other industrial applications, yet due to the recalcitrant nature of plastics, the development here has been limited. However, the improvement could have substantial applications in various sectors as incorporating enzymes into plastic degradation processes makes the process more sustainable and energy efficient compared to conventional mechanical recycling. This method of recycling applies force to break down plastic polymer length, compromising the polymer’s properties and restricting its potential applications. This practice, often termed ‘downcycling’, does not constitute true recycling, yielding products with limited utility in each iteration [24]. However, innovation is being driven by start-up enterprises like Carbios who are committed to testing a potentially ground-breaking technology aimed at establishing the world’s first plastics recycling facility that relies entirely on enzymatic recycling. This technology could revolutionize plastic waste facilities, yet several challenges must be addressed before this becomes a cost-effective process.

One major challenge is the need to engineer enzymes that can efficiently degrade a wide range of plastics beyond PET. Different plastic polymers possess distinct chemical structures and require specific enzymatic activities for degradation, yet a sample of plastic waste can contain a whole range of polymers. Therefore, efforts are underway to identify organisms and enzymes capable of targeting plastics other than PET, such as PS [25], PVC [26] and PE [27] yet the research effort is considerably lower than that of PET.

Another challenge lies in the scaling and cost-effectiveness of enzyme production. The industrial-scale production of enzymes requires the optimization of fermentation processes and the development of robust enzyme expression systems. Additionally, the cost of enzyme production must be competitive with traditional plastic waste management methods to enable significant adoption. Furthermore, the compatibility of plastic-degrading enzymes with existing plastic recycling infrastructure needs to be considered. Enzymes should not interfere with the quality of recycled plastic materials or introduce undesirable contaminants. Ensuring compatibility between enzyme-based degradation processes and existing recycling technologies is vital for the integration of enzymes into the plastic recycling process.

If scientists and enterprises like Carbios can create a successful technology the potential could herald a revolutionary step towards the development of a circular economy. However, policymakers will require cost-effective methods before the widespread implementation of plastic-degrading enzymes.

#### 2.2.3. Increase Efficiency under Environmental Conditions

The problem of plastic waste is growing, and sadly, a significant proportion of the environmental burden arises not only from plastics that are properly recycled but from those that find their way into natural ecosystems. Raising the temperature in situ is, therefore, not a practical solution for tackling plastics outside of specialist facilities, despite the clear advantage that it increases the proportion of amorphous PET. Moreover, finding alternative methods to high temperatures used in industrial processes would decrease operational expenses and offer a more environmentally friendly solution to recycling as well.

There are ongoing investigations into mesophilic organisms for their ability to operate optimally at lower temperatures [28]. However, these studies seem to focus on plastics other than PET, such as PBAT, which has lower strength and is more amorphous, making it easier to degrade at lower temperatures. More research would be needed here to address the more crystalline plastics.

In a study conducted in 2022, the objective was to engineer the hydrolytic activity of PETase at moderate temperatures, offering an alternative approach to developing a thermally stable enzyme [9]. They achieved this by developing a neural network to conduct a comprehensive in silico mutagenesis followed by extensive experimental validation and characterization. Five mutations (S132E, D186H, R224Q, N233K and R280A) were identified that not only showed improved kinetics but were highly functional at 50 °C. The resulting enzymes with these five mutations, known as FAST-PETase, exhibited a 29-fold increase in hydrolysis of amorphous PET at 40 °C compared to the wild-type PET enzyme. Analysis of crystal structures of FAST-PETase at a resolution of 1.44 Å allowed elucidation of the effect of these mutations on enhancing thermostability. For instance, the replacement of Asp with a positively charged Lys at position 233 facilitates the formation of a salt bridge with glutamate at position 204. Additionally, R224Q and S132E were observed to promote increased hydrogen bonding within the FAST-PETase. This study illustrates how large amounts of data can undergo computational analysis to identify where alternative bonds can be created. This allows the increase in thermostability, as seen in FAST-PETase, that would be difficult to achieve through experimental testing alone.

However, a significant challenge when researching and engineering these enzymes for industrial or environmental use lies in the extensive range of crystallinity present in plastic waste. Current experimental set ups in the existing literature often use standardized PET with specific crystallinity levels that are lower than the diverse crystallinity found in real-world plastic waste. Many studies initially validate their findings using only amorphous PET, which is not representative of the plastic problem, thus limiting the applicability of the results to real-world scenarios.

When assessing these engineered enzymes, it is crucial to test them on samples intended for recycling. The study investigating FAST-PETase used a variety of post-consumer plastic waste samples with controlled variations in crystallinity and compared the results with other thermotolerant PETase mutants in addition to the wild-type [9]. Their study revealed that FAST-PETase exhibited the highest efficiency at 50 °C across the range of samples but at temperatures above 60 °C, the more thermally stable mutant PETases performed better. None of the enzymes, however, were able to completely degrade the plastic sample with the highest crystallinity (>25% crystallinity) even though the researchers have clearly improved the FAST-PETase rate of PET degradation rate at 50 °C. The potential of deployment of these enzymes environmentally is, therefore, unlikely until the inability to degrade the crystalline polymer is addressed. Further research showed that when PET underwent a thermal pre-treatment method, the FAST-PETase could completely degrade the plastic regardless of the initial crystallinity. This shows how PET and other plastic degrading enzymes could be highly useful within industrial recycling plants but not in the broader environment.

#### 2.2.4. Overcoming the Hydrophobicity of Plastics and PET to Increase Efficiency

The recalcitrant nature of many plastics can be attributed to their hydrophobicity and insolubility which heavily reduces their biodegradability. Considerations to reduce the impact of this in waste management include Ultraviolet irradiation to increase the number of hydroxyl groups thus improving the surface properties for bio degradation [2]. For other enzymes, the initial adsorption is presumably mediated by interactions with the hydrophobic regions surrounding the active site. An esterase reported to hydrolyse PBAT was engineered to increase the rate of degradation through deletion of the N-terminus [29]. This exposed a hydrophobic patch on the enzyme that was proposed to increase the attraction of the secreted enzyme to the PBAT and could be implemented to other enzymes (Figure 4A).

An alternative approach under consideration is from the cellulase research community where the multi-modular architecture of cellulases, consisting of both a catalytic module and a carbohydrate-binding module, is well established. This modular design creates a synergy that significantly increases the catalytic efficiency of these enzymes [30]. This has highlighted a potential avenue for engineering higher-affinity accessory modules for PETases, which could increase their local concentration at the PET surface and thereby enhance their efficiency (Figure 4B). Designing a PET-binding module de novo remains a significant challenge due to the limitations of predicting how protein sequences govern function [31]. A better approach may be to look to nature, which is regarded as the world’s ‘best protein engineer’, and identify a protein with a natural affinity for PET.

Other multi-modular systems are also being investigated. It has been demonstrated that MHETase, the second of the pair of PET-degrading enzymes from *I. sakaiensis,* shares a similar structure with PETase. As expected, the degradation of plastic using both enzymes was faster compared to PETase alone and no activity was observed for MHETase alone. Excitingly, the researchers went on to create a chimeric PET-MHET dual enzyme, which improved the hydrolysis rate by more than threefold [32]. This suggests a synergistic effect resulting from the proximity of these consecutive enzymes (Figure 4C).

Interestingly, the cellulase research community is conducting extensive research on the cellulosome, an extracellular multi-enzyme complex that significantly enhances cellulose degradation [33]. This research holds promise for enhancing industrial processes involving many enzymes. Therefore, parallels can also be drawn with the synergistic effect of the PET-MHET dual enzymes and may be a useful concept for rate improvement and future application of plastic-degrading enzymes.

#### 2.2.5. A More Collaborative Global Approach for the Future Plastic Management

The concept of a circular closed economy is often discussed as the objective to work towards the future. Governments and advocates have perpetuated the idea of a circular economy, inspiring scientists to develop sustainable solutions for plastic management. Start-ups like Carbios are making paths for technological development but other organizations are looking to collaborate with the global picture in mind. Revolution Plastics, based at the University of Portsmouth (UK) [34], is an organization looking to make a global impact. This pioneering initiative serves as a platform to unite teams of researchers, business leaders, campaigners and citizens, all working towards advancing the shift to a more circular economic model. Noteworthy components within Revolution Plastics encompass a Microplastics Research Group, an Advanced Materials and Manufacturing Research Group, and a Centre for Enzyme Evolution. Importantly the various research groups are interconnected through a global plastics policy centre, ensuring the conversion of research into beneficial outcomes to the society.

A groundbreaking announcement was made in May 2023 by the Intergovernmental Negotiating Committee, revealing plans for an innovative plastics treaty set to be enforced by 2025 [35]. This represents a pivotal advancement, yet its success will depend on extensive collaboration among researchers, policymakers, industry leaders and environmental organizations. Initiatives of this calibre provide a glimmer of hope for a future where plastics can be utilized sustainably, without causing harm to the environment.

## 3. Future Directions and Conclusions

In this review. we began by assessing our current understanding of how PETase functions and we found ourselves both surprised and perplexed by the apparent evolutionary deviations that have set these enzymes apart from their cutinase counterparts. PETase has clearly retained the overall α/β hydrolases structure yet subtle adjustments, like the ‘wobbling’ Trp residue, have occurred to accommodate the PET substrates. However, multiple studies show that this can be pushed further to create faster variants than the wild type and as our grasp of how altering PETase potentially improves the rate, we could apply this to tackle a spectrum of enzymes. The rise of computation and AI models may also help us uncover the patterns that nature seems to follow yet improvements may rely on directed evolution to truly enhance these enzymes’ efficiency.

Over the past decade, efforts to create an enzyme with the capacity to degrade crystalline PET without the need for additional pre-treatment and temperature increase have proven futile. While this may be of lesser concern within industrial applications, the question arises: Can an enzyme be effectively deployed in natural ecosystems to counter the growing impact of microplastics? A clearer approach may involve establishing proper tracing and accountability mechanisms for plastic production while simultaneously racing to discover more effective enzymatic solutions within industrial waste management. The success of such methods would alleviate not only environmental pollution but also the potential unintended consequences and ethical assessments necessary before using these enzymes in natural ecosystems.

However, effectively addressing plastic waste comprehensively demands the exploration of various enzymes for the biodegradation of other synthetic polyesters like nylon and PVC. Understanding the molecular intricacies of the PETases has been a first step in harnessing the potential enzymes for plastic degradation yet other more hydrophobic plastics may provide a greater challenge. Combining PETase with other plastic-binding domains or even creating multi-enzyme complexes could alleviate some of the pressure on solely enhancing the PETase itself to increase the degradation rate. Instead, these methods could focus on enhancing the degradation processes, alternatively to just the PETase mechanism. Drawing inspiration from other fields where nature has developed techniques to break down resilient polymers, such as cellulose, might provide valuable insights. Given that plastics are relatively new materials, having been mass manufactured only in the past century, natural evolution has not had sufficient time to develop its own solutions for their decomposition. Intervening and utilizing existing mechanisms could potentially offer the necessary solution to address this challenge.

A profound level of collaboration involving researchers, policymakers and environmental organizations will play a pivotal role in guaranteeing the allocation of funding for pertinent research and facilitating innovation in this domain. The prospect of engineering enzymes to address our waste presents an alluring opportunity that could significantly contribute to safeguarding our planet. However, the obstacles of efficiency, scalability and versatility remain unresolved challenges that must be overcome before the issue escalates to an irreversible place.

## Figures and Tables

**Figure 1 biomolecules-13-01407-f001:**
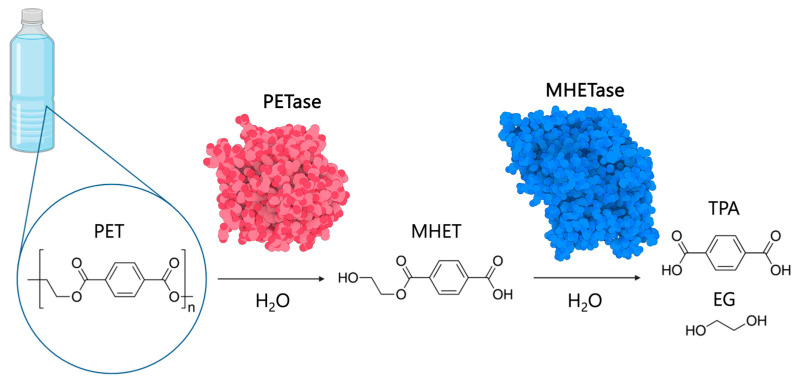
Sequential action of PETase and MHETase, isolated from *I. sakaiensis*, that degrade PET into products for metabolism. PET (polyethylene terephthalate), MHET (mono (2-hydroxyethyl) terephthalic acid), terephthalate (TPA) and EG (ethylene glycol). Own figure created using Biorender.com (accessed 18 April 2023) and 3D Protein Imaging software [7] (accessed 20 April 2023) using input from the Protein Data Bank (PDB). The PDB codes used are: (6EQD—PETase) and (6QZ1—MHETase).

**Figure 2 biomolecules-13-01407-f002:**
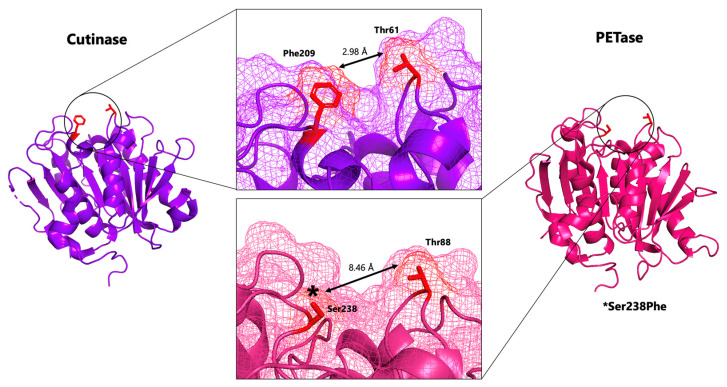
Comparison of substrate binding cleft between Cutinase (orange) and PETase (blue) in relation to overall structure which contains the α/β-hydrolase structural fold. * Marks Ser238Phe mutation that caused narrowing of channel in PETase yet did not decrease enzyme activity. Figure adapted from Austin et al. [8]. Figure created using PyMol. PDB codes used: (PETase: 6EQD, where first residue M is position 2 throughout the review) and (Cutinase: 4CG1).

**Figure 3 biomolecules-13-01407-f003:**
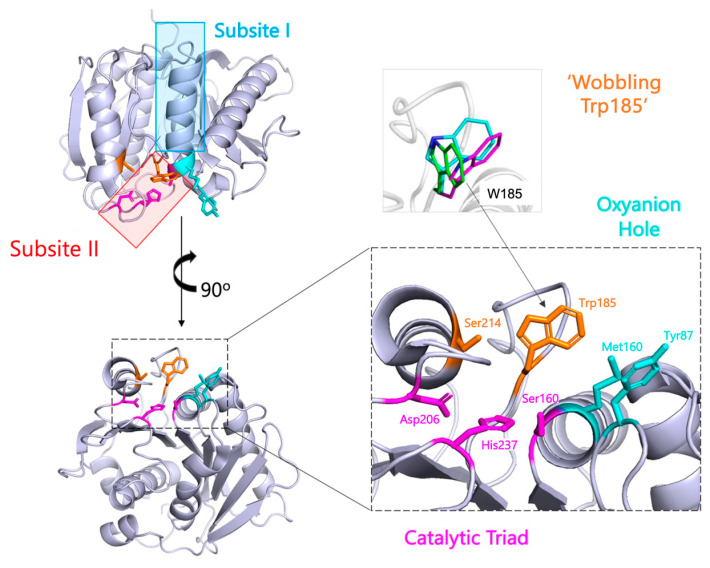
Summary of structural knowledge gleaned from X-ray crystallography and molecular docking simulations on the PETase, shown as grey in figure. Subsites I and Subsite II identified by Joo et al. [12] are labelled in Red and Blue, respectively. Identification of Catalytic triad (pink), the oxyanion hole (cyan) and the wobbling W156 permitted by the S185 mutation (orange) identified in the PETase. Different structural conformations of W156 were taken from Han et al. [13]. Figure created using PyMol (PDB: 6EQD).

**Figure 4 biomolecules-13-01407-f004:**
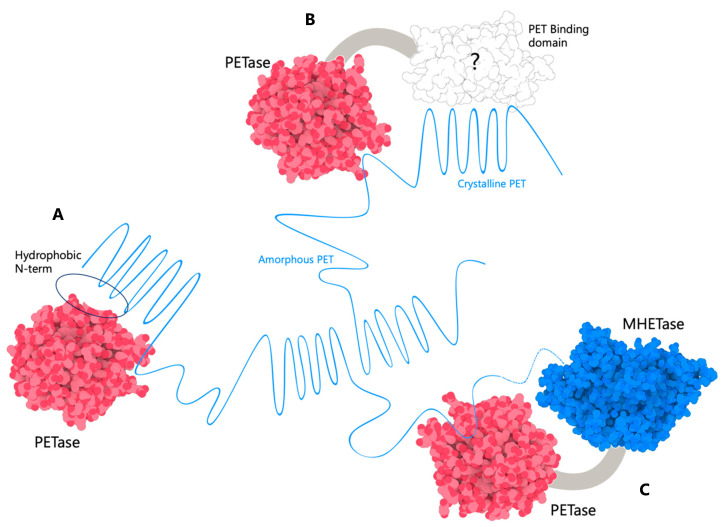
Methods under investigation to increase PET degradation efficiency and reduce PET hydrophobicity. (**A**) An exposed hydrophobic patch on the enzyme N-terminus which may increase the attraction of the enzyme to PET, (**B**) the possibility of creating a PET-binding accessory domain, marked by (?), for PET which could increase local concentration at the PET surface and thereby enhance their efficiency, and (**C**) chimeric PET-MHET dual enzyme. Illustration was created using 3D Protein Imaging software [7] (accessed: 20 July 2023) PDB: (6EQD—PETase) and (6QZ1—MHETase).

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
