# Peer review of "Engineering Plastic Eating Enzymes Using Structural Biology"

_biomolecules, 2023, doi:10.3390/biom13091407_

Round 1

Reviewer 1 Report

The review ‘Engineering Plastic Eating Enzymes using Structural Biology’ by Barclay and Acharya, recapitulates the problems arising from the enormous amount of plastics that has to be somehow degraded in order to mitigate its negative effects on the environment.

The paper examines the current biochemical and biophysical methods that may facilitate the development of enzymes capable of degrading PET, one of the most extensively used plastics. It then discusses the challenges that have to be addressed for achieving substantial advancements in using these enzymes as a solution to the plastic pollution problem.

The focus of the paper is an enzyme (isolated from the bacterium Ideonella sakaiensis) capable of carrying out the first of the two steps that are necessary to hydrolize PET (the second step being carried out by MHETase). The catalytic mechanism of the enzyme is sketched and possible strategies to improve its activity, thermostability and efficiency are proposed based on an accurate comparative analysis of existing crystal structures of such enzyme and homologous ones.

Also, the last part concerning future directions is pertinent and it indicates fruitful avenues to follow for the optimization of such enzymes and for a better way to deal with pollution problems (an integrated approach amongst several kinds of institutions).

The paper is probably a bit concise for a review, but it is well structured, smoothly readable and understandable by a wide audience. The approach of the structural analysis is sound as well as the considerations outlined. The crystal structure presented has been and well analyzed/compared with other homologues structures.

Literature appears to be adequately cited related to the length of the review.

I have just one minor point/request and a curiosity:

I think the review would benefit from a little table/graph showing the turnover number, or Kcat or Michaelis-Menten constant for the enzyme analyzed/engineered if those numbers are available. Thus, I think the authors should include it in the paper to give the reader an immediate view of the comparison of the possible efficiency of PETase mutants.

Authors also mention at least two private companies dealing with the specific issue they deal with in the paper. Have they interacted with them directly? This is maybe just a curiosity but nevertheless…

Typos:

Page 1, line 38 and 45, page 2 line 59, page 3 line 101: I guess ‘cutinises’ should be ‘cutinases’ in all cases throughout the paper.

Page 4, line 122: Should ‘PETase’ be ‘PETs’?

Page 7, line 291: ‘Further research showed that when PETase underwent a thermal pre-treatment method all crystallinity of PET could be completely degraded that enabled complete degraded’

Something is not clear in the last part of this sentence and the full stop is missing at the end of the sentence itself.

Reviewer 2 Report

The article attempts to review a current state and perspective direction in the actual field of engineering of Plastic digesting enzymes, especially PET (polyethylene terephthalate) hydrolysing enzymes.

As a not specialist in this field, for me it was an interesting possibility to get insight into the role of structural methods in solving actual and interesting problems concerning the whole society. I would comment more on the structural part of this paper. Several points are simply typos, also a structural part needs in my opinion some additional work on text.

Introduction
Lines 38 and 45, 59, 101: cutinises , maybe cutinases?

2.1.1. Efforts to Understand Catalytic Mechanism Showed Engineering Could
Enhance Rate
In this chapter the case of improvement of the catalytic activity of
PETase by single mutation Ser238Phe is discussed based on [7]. In
[7] is also described S238F/W159H double mutant and W185A mutant. Should these mutants be mentioned here too? From the hand, in my opinion, this case would better be placed into the second chapter (2.2. Engineering the Enzyme to Enhance Efficiency), that deals with numerous attempts to improve natural enzymes.

Would it be more logical to rename this chapter to something like "2.1.1. Catalytic Mechanism" and describe the active center of enzyme, catalytic triad (Ser-Asp-His) shared by cutinases and PETases, whether there are
variants, substitutions etc.

It is also possible to rename the chapter to the "PETase is similar to
Cutinase" or "Comparison of ...", and comment on differences and similarities
between cutinases and PETases including the variations of Ser/Phe
residues at the substrate-binding cleft. For example, authors can
finish the chapter on Line 106, and the text starting from the sentence "Austin et al [7] hypothesized..." transfer to the Section 2 and there further
describe the enhancement of PETase activity by mutations of the cleft
residues to cutinase residues. Also, in ref. [7, Austin2018] as far as
I understand is described double mutant S238F/W159H with enhanced
ability to hydrolyse PET by better substrate binding. Such comparison
could be used in the following Section "Engineering..."

Line 116: Ser234Phe should be Ser238Phe?

Figure 2.

In the Legend "Key α/β-hydrolase structural fold labelled."
It is not labelled. The complete structure is colored blue. In order
to show fold, one can color fold-specific secondary structure elements
excluding unspecific loops. But in this case it is maybe not so
important, as enzymes from this group are rather similar (see
structural comparison in [12, Han2017].

2.1.2.
Efforts to Understand the Enzyme-Substrate Complex

In the absence of crystal structure MD simulations is quite the method
of choice over the last decades. The crystallization of mutant (inactivated) enzyme is also not so bad. For example, maybe worth mention crystals with substrate obtained using inactive mutant S165A in [9, Tornier2020]. But this study was performed with Leaf-branch compost cutinase (LCC), enzyme related to IsaPETase. But probably, cutinases are out of the scope in this review.

Along with the example presented in paper, maybe it is worth also mention the already cited ref. 7 [Austin2018] where the MD simulations were also used for modeling of two substrates (PET and PEF) binding by wt PETase and S238F/W159H double mutant?

In [11, Joo2018] catalytic triad is formed by Ser160, His237, and
Asp206, and oxyanion hole consists of nitrogen atoms of Tyr87 and
Met160. In current review corresponding catalytic residues are cited as
Ser131, His208, and Asp177. The numbering differ by 29 residues,
probably corresponding to His-tag in the Joo2018. It is OK, but "...Tyr87
and Met160, which stabilizes the tetrahedral intermediate..." are the
same numbers as in [11]. MAybe it corresponds to Y58 and M132 like in
reference [12]? Authors should keep the consistent numbering
throughout the whole text and all Figures and, maybe, mention which
numbering is used (as in PDB file, or according to the protein
sequence without any tags).

In [11, Joo2018] catalytic triad is formed by Ser160, His237, and
Asp206, and oxyanion hole consists of nitrogen atoms of Tyr87 and
Met160. In current review corresponding catalytic residues are cited as
Ser131, His208, and Asp177. The numbering differ by 29 residues,
probably corresponding to His-tag in the Joo2018. It is OK, but "...Tyr87
and Met160, which stabilizes the tetrahedral intermediate..." are the
same numbers as in [11]. Maybe it corresponds to Y58 and M132 like in
reference [12]? Authors should keep the consistent numbering
throughout the whole text and all Figures and, maybe, mention which
numbering is used (as in PDB file, or according to the protein
sequence without any tags).

2.1.3.
Identification of Unique Enzyme Feature – Wobbling Trp

Trp156 is discussed here, that is wobbling between several crystal structures
at the entrance to substrate binding cleft (subsite I). This could be a part of a chapter describing substrate binding.

[15, Zeng2022] is cited along with the sentence "...Ser185 is responsible for accommodating the range of conformations of Trp156, which the larger His residue would not permit" But, firstly in [15] is dealing with ICCP protein from plant leaf-branch compost cutinase (PDB 7VVC) mutant, IsaPetase. Maybe better to cite [14 etc.]?

Line 150: "... it adopts a fixed C conformer [14]." Maybe to
reformulate this sentence to something more descriptive, as "C
conformer" looks strange. And should not the reference be change to
[12, Han2017]? In [14, Fecker2018] authors found out that this Trp is not fixed well in their structure. Three distinct conformations were found in structure from ref. [12, Han2017], where authors compared several available homologous structures with PETase and found out that in homologous structures Trp adopts a more or less single conformation, that is unfavorable for substrate binding (C conformer in their structure).

Line 157: Trp156 is mentioned as Trp165 (correct?)

2.2.1.
Developing a Thermostable Enzyme to Enhance Efficiency

Lines 182-184:
"Since PETase currently operates within a narrow pH range and is
resistant to temperature change, significant research efforts are
currently focused on the formidable task of developing a highly
thermally stable PETase." Are there references to pH range and
resistance to temperature changes?

Page 6, first paragraph

In the comprehensive study [9, Tournier2020] indeed, are described the
redesign of Me-binding into Cys-Cys bridge by mutation
Asp238Cys/Ser283C and all the other mutations improving catalytic
activity as well as thermostability. But I believe it should be
mentioned that the study deals with leaf-branch compost cutinase (LCC)
[9, Tournier2020]. So, strictly speaking, it is not the pETase,
albeit closely related to PETase from Ideonella sakaiensis. Maybe,
authors should mention this, if cutinases are within the scope of this
paper. Also, ICC variant with improved activity is mentioned, but
maybe it is worth discuss also the mutants with increased
thermostability (ICCG, ICCM etc.). And maybe add more on results from
ref. [15], that continues research of most effective CCL variant
(ICCG). Especially if the subsection is about thermostability.

Line 192: Correct mutation Asp238Cys/Ser283 to Asp238Cys/Ser283Cys

2.2.2.
Industrial Applications and Challenges

Line 282:
"...PETase mutants (ICCM and LCC) in addition to the wild- type [8]." Here, as far as I understand, is a comment on experiment presented in Figure 3 in ref. [8], where FAST-PETase created based on Ideonella sakaiensis PETase is compared with LCC mutants (ICCM) and wt as well as with commercial mutants of Ideonella sakaiensis PETase (Thermo-PETase and DURA-PETase).

References
Refs 5, 7, 10-12 17, 20, 24-27 pages are absent
